# INPUT COMPLEXITY AND OUT-OF-DISTRIBUTION DETECTION WITH LIKELIHOOD-BASED GENERATIVE MODELS

**Joan Serrà**[1,2]**, David Álvarez**[2,3]**, Vicenç Gómez**[4]**, Olga Slizovskaia**[2,4]**, José F. Núñez**[4]**, and Jordi Luque**[2]

[1] Dolby Laboratories, Barcelona, Spain

[2] Telefónica Research, Barcelona, Spain

[3] Universitat Politècnica de Catalunya, Barcelona, Spain

[4] Universitat Pompeu Fabra, Barcelona, Spain

`joan.serra@dolby.com,vicen.gomez@upf.edu`

## ABSTRACT

Likelihood-based generative models are a promising resource to detect out-of-distribution (OOD) inputs which could compromise the robustness or reliability of a machine learning system. However, likelihoods derived from such models have been shown to be problematic for detecting certain types of inputs that significantly differ from training data. In this paper, we pose that this problem is due to the excessive influence that input complexity has in generative models' likelihoods. We report a set of experiments supporting this hypothesis, and use an estimate of input complexity to derive an efficient and parameter-free OOD score, which can be seen as a likelihood-ratio, akin to Bayesian model comparison. We find such score to perform comparably to, or even better than, existing OOD detection approaches under a wide range of data sets, models, model sizes, and complexity estimates.

## 1 INTRODUCTION

Assessing whether input data is novel or significantly different than the one used in training is critical for real-world machine learning applications. Such data are known as out-of-distribution (OOD) inputs, and detecting them should facilitate safe and reliable model operation. This is particularly necessary for deep neural network classifiers, which can be easily fooled by OOD data (Nguyen et al., 2015). Several approaches have been proposed for OOD detection on top of or within a neural network classifier (Hendrycks & Gimpel, 2017; Lakshminarayanan et al., 2017; Liang et al., 2018; Lee et al., 2018). Nonetheless, OOD detection is not limited to classification tasks nor to labeled data sets. Two examples of that are novelty detection from an unlabeled data set and next-frame prediction from video sequences.

A rather obvious strategy to perform OOD detection in the absence of labels (and even in the presence of them) is to learn a density model $\mathcal{M}$ that approximates the true distribution $p^*(\mathcal{X})$ of training inputs $\mathbf{x} \in \mathcal{X}$ (Bishop, 1994). Then, if such approximation is good enough, that is, $p(\mathbf{x}|\mathcal{M}) \approx p^*(\mathbf{x})$, OOD inputs should yield a low likelihood under model $\mathcal{M}$. With complex data like audio or images, this strategy was long thought to be unattainable due to the difficulty of learning a sufficiently good model. However, with current approaches, we start having generative models that are able to learn good approximations of the density conveyed by those complex data. Autoregressive and invertible models such as PixelCNN++ (Salimans et al., 2017) and Glow (Kingma & Dhariwal, 2018) perform well in this regard and, in addition, can approximate $p(\mathbf{x}|\mathcal{M})$ with arbitrary accuracy.

Recent works, however, have shown that likelihoods derived from generative models fail to distinguish between training data and some OOD input types (Choi et al., 2018; Nalisnick et al., 2019a; Hendrycks et al., 2019). This occurs for different likelihood-based generative models, and even when inputs are unrelated to training data or have totally different semantics. For instance, when

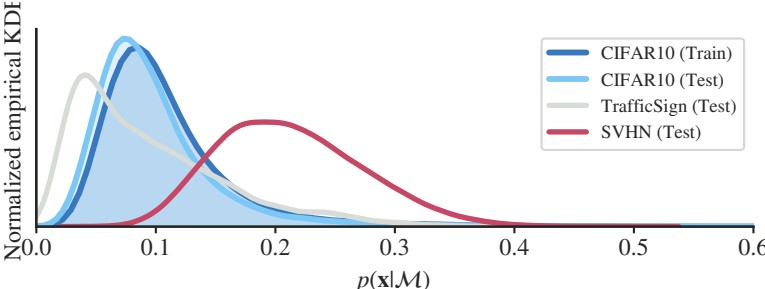

Figure 1: Likelihoods from a Glow model trained on CIFAR10. Qualitatively similar results are obtained for other generative models and data sets (see also results in Choi et al., 2018; Nalisnick et al., 2019a).

trained on CIFAR10, generative models report higher likelihoods for SVHN than for CIFAR10 itself (Fig. 1; data descriptions are available in Appendix A). Intriguingly, this behavior is not consistent across data sets, as other ones correctly tend to produce likelihoods lower than the ones of the training data (see the example of TrafficSign in Fig. 1). A number of explanations have been suggested for the root cause of this behavior (Choi et al., 2018; Nalisnick et al., 2019a; Ren et al., 2019) but, to date, a full understanding of the phenomenon remains elusive.

In this paper, we shed light to the above phenomenon, showing that likelihoods computed from generative models exhibit a strong bias towards the complexity of the corresponding inputs. We find that qualitatively complex images tend to produce the lowest likelihoods, and that simple images always yield the highest ones. In fact, we show a clear negative correlation between quantitative estimates of complexity and the likelihood of generative models. In the second part of the paper, we propose to leverage such estimates of complexity to detect OOD inputs. To do so, we introduce a widely-applicable OOD score for individual inputs that corresponds, conceptually, to a likelihood-ratio test statistic. We show that such score turns likelihood-based generative models into practical and effective OOD detectors, with performances comparable to, or even better than the state-of-the-art. We base our experiments on an extensive collection of alternatives, including a pool of 12 data sets, two conceptually-different generative models, increasing model sizes, and three variants of complexity estimates.

## 2 COMPLEXITY BIAS IN LIKELIHOOD-BASED GENERATIVE MODELS

From now on, we shall consider the log-likelihood of an input $\mathbf{x}$ given a model $\mathcal{M}$: $\ell_{\mathcal{M}}(\mathbf{x}) = \log_2 p(\mathbf{x}|\mathcal{M})$. Following common practice in evaluating generative models, negative log-likelihoods $-\ell_{\mathcal{M}}$ will be expressed in bits per dimension (Theis et al., 2016), where dimension corresponds to the total size of $\mathbf{x}$ (we resize all images to $3{\times}32{\times}32$ pixels). Note that the qualitative behavior of log-likelihoods is the same as likelihoods: ideally, OOD inputs should have a low $\ell_{\mathcal{M}}$, while in-distribution data should have a larger $\ell_{\mathcal{M}}$.

Most literature compares likelihoods of a given model for a few data sets. However, if we consider several different data sets at once and study their likelihoods, we can get some insight. In Fig. 2, we show the log-likelihood distributions for the considered data sets (Appendix A), computed with a Glow model trained on CIFAR10. We observe that the data set with a higher log-likelihood is Constant, a data set of constant-color images, followed by Omniglot, MNIST, and FashionMNIST; all of those featuring gray-scale images with a large presence of empty black background. On the other side of the spectrum, we observe that the data set with a lower log-likelihood is Noise, a data set of uniform random images, followed by TrafficSign and TinyImageNet; both featuring colorful images with non-trivial background. Such ordering is perhaps more clear by looking at the average log-likelihood of each data set (Appendix D). If we think about the visual complexity of the images in those data sets, it would seem that log-likelihoods tend to grow when images become simpler and with less information or content.

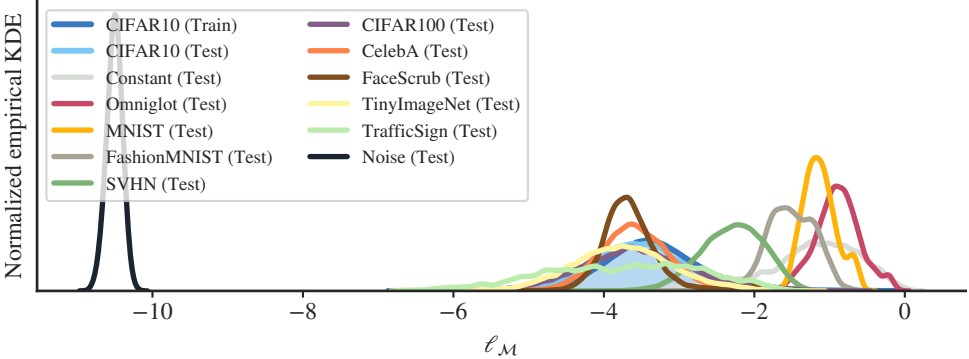

Figure 2: Log-likelihoods from a Glow model trained on CIFAR10. Qualitatively similar results are obtained for a PixelCNN++ model and when training with FashionMNIST.

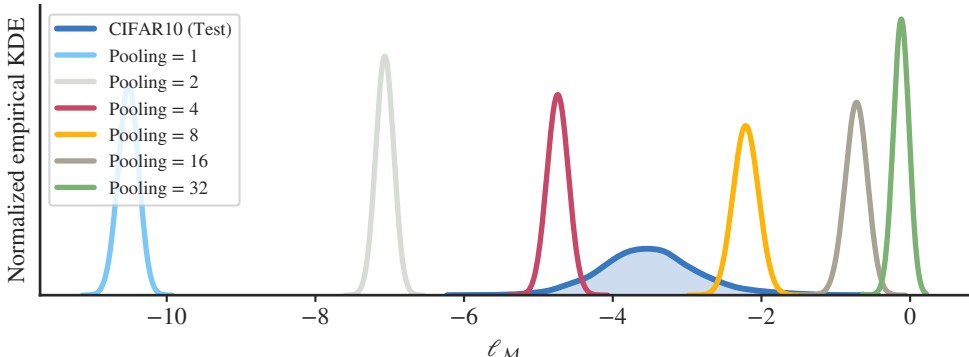

Figure 3: Pooled-image log-likelihoods obtained from a Glow model trained on CIFAR10. Qualitatively similar results are obtained for a PixelCNN++ model.

To further confirm the previous observation, we design a controlled experiment where we can set different decreasing levels of image complexity. We train a generative model with some data set, as before, but now compute likelihoods of progressively simpler inputs. Such inputs are obtained by average-pooling the uniform random Noise images by factors of 1, 2, 4, 8, 16, and 32, and re-scaling back the images to the original size by nearest-neighbor up-sampling. Intuitively, a noise image with a pooling size of 1 (no pooling) has the highest complexity, while a noise image with a pooling of 32 (constant-color image) has the lowest complexity. Pooling factors from 2 to 16 then account for intermediate, decreasing levels of complexity. The result of the experiment is a progressive growing of the log-likelihood $\ell_{\mathcal{M}}$ (Fig. 3). Given that the only difference between data is the pooling factor, we can infer that image complexity plays a major role in generative models' likelihoods.

Until now, we have consciously avoided a quantitative definition of complexity. However, to further study the observed phenomenon, and despite the difficulty in quantifying the multiple aspects that affect the complexity of an input (cf. Lloyd, 2001), we have to adopt one. A sensible choice would be to exploit the notion of Kolmogorov complexity (Kolmogorov, 1963) which, unfortunately, is non-computable. In such cases, we have to deal with it by calculating an upper bound using a lossless compression algorithm (Cover & Thomas, 2006). Given a set of inputs $\mathbf{x}$ coded with the same bit depth, the normalized size of their compressed versions, $L(\mathbf{x})$ (in bits per dimension), can be considered a reasonable estimate of their complexity. That is, given the same coding depth, a highly complex input will require more bits per dimension, while a less complex one will be compressed with fewer bits per dimension. For images, we can use PNG, JPEG2000, or FLIF compressors (Appendix C). For other data types such as audio or text, other lossless compressors should be available to produce a similar estimate.

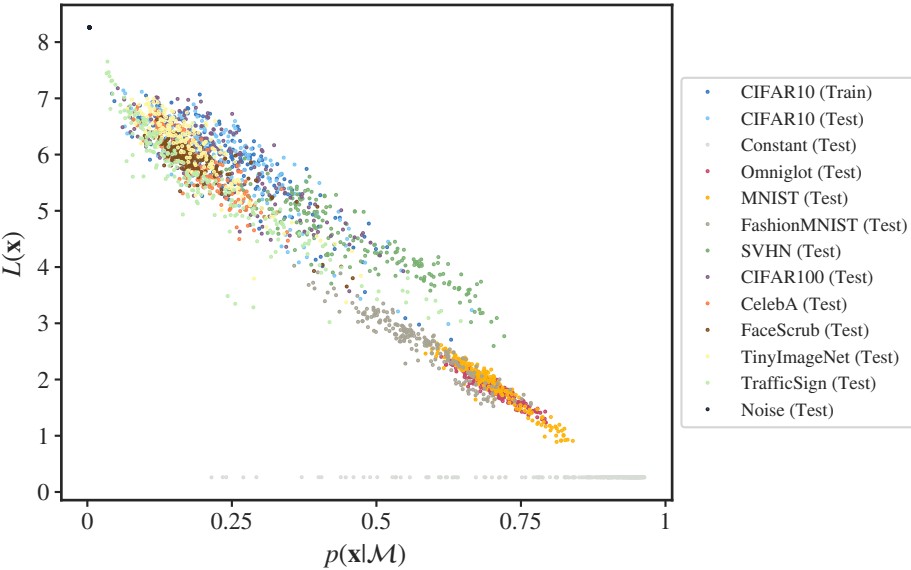

Figure 4: Normalized compressed lengths using a PNG compressor with respect to likelihoods of a PixelCNN++ model trained on CIFAR10 (for visualization purposes we here employ a sample of 200 images per data set). Similar results are obtained for a Glow model and other compressors.

If we study the relation between generative models' likelihoods and our complexity estimates, we observe that there is a clear negative correlation (Fig. 4). Considering all data sets, we find Pearson's correlation coefficients below $-0.75$ for models trained on FashionMNIST, and below $-0.9$ for models trained on CIFAR10, independently of the compressor used (Appendix D). Such significant correlations, all of them with infinitesimal p-values, indicate that likelihood-based measures are highly influenced by the complexity of the input image, and that this concept accounts for most of their variance. In fact, such strong correlations suggest that one may replace the computed likelihood values for the negative of the complexity estimate and obtain almost the same result (Appendix D). This implies that, in terms of detecting OOD inputs, a complexity estimate would perform as well (or bad) as the likelihoods computed from our generative models.

## 3 TESTING OUT-OF-DISTRIBUTION INPUTS

### 3.1 DEFINITION

As complexity seems to account for most of the variability in generative models' likelihoods, we propose to compensate for it when testing for possible OOD inputs. Given that both negative log-likelihoods $-\ell_{\mathcal{M}}(\mathbf{x})$ and the complexity estimate $L(\mathbf{x})$ are expressed in bits per dimension (Sec. 2), we can express our OOD score as a subtraction between the two:

$$S(\mathbf{x}) = -\ell_{\mathcal{M}}(\mathbf{x}) - L(\mathbf{x}). \tag{1}$$

Notice that, since we use negative log-likelihoods, the higher the $S$, the more OOD the input $\mathbf{x}$ will be (see below).

### 3.2 INTERPRETATION: OCCAM'S RAZOR AND THE OUT-OF-DISTRIBUTION PROBLEM

Interestingly, $S$ can be interpreted as a likelihood-ratio test statistic. For that, we take the point of view of Bayesian model comparison or minimum description length principle (MacKay, 2003). We can think of a compressor $\mathcal{M}_0$ as a universal model, adjusted for all possible inputs and general enough so that it is not biased towards a particular type of data semantics. Considering the probabilistic model associated with the size of the output produced by the lossless compressor, we

have

$$p(\mathbf{x}|\mathcal{M}_0) = 2^{-L(\mathbf{x})}$$

and, correspondingly,

$$L(\mathbf{x}) = -\log_2 p(\mathbf{x}|\mathcal{M}_0). \tag{2}$$

In Bayesian model comparison, we are interested in comparing the posterior probabilities of different models in light of data $\mathcal{X}$. In our setting, the trained generative model $\mathcal{M}$ is a 'simpler' version of the universal model $\mathcal{M}_0$, targeted to a specific semantics or data type. With it, one aims to approximate the marginal likelihood (or model evidence) for $\mathbf{x} \in \mathcal{X}$, which integrates out all model parameters:

$$p(\mathbf{x}|\mathcal{M}) = \int p(\mathbf{x}|\theta, \mathcal{M})p(\theta|\mathcal{M})d\theta.$$

This integral is intractable, but current generative models can approximate $p(\mathbf{x}|\mathcal{M})$ with arbitrary accuracy (Kingma & Dhariwal, 2018). Choosing between one or another model is then reduced to a simple likelihood ratio:

$$\log_2 \frac{p(\mathcal{M}_0|\mathbf{x})}{p(\mathcal{M}|\mathbf{x})} = \log_2 \frac{p(\mathbf{x}|\mathcal{M}_0)p(\mathcal{M}_0)}{p(\mathbf{x}|\mathcal{M})p(\mathcal{M})}. \tag{3}$$

For uniform priors $p(\mathcal{M}_0) = p(\mathcal{M}) = 1/2$, this ratio is reduced to

$$S(\mathbf{x}) = -\log_2 p(\mathbf{x}|\mathcal{M}) + \log_2 p(\mathbf{x}|\mathcal{M}_0)$$

which, using Eq. 2 for the last term, becomes Eq. 1.

The ratio $S$ accommodates the Occam's razor principle. Consider simple inputs that can be easily compressed by $\mathcal{M}_0$ using a few bits, and that are not present in the training of $\mathcal{M}$. These cases have a high probability under $\mathcal{M}_0$, effectively correcting the abnormal high likelihood given by the learned model $\mathcal{M}$. The same effect will occur with complex inputs that are not present in the training data. In these cases, both likelihoods will be low, but the universal lossless compressor $\mathcal{M}_0$ will predict those better than the learned model $\mathcal{M}$. The two situations will lead to large values of $S$. In contrast, inputs that belong to the data used to train the generative model $\mathcal{M}$ will always be better predicted by $\mathcal{M}$ than by $\mathcal{M}_0$, resulting in lower values of $S$.

### 3.3 USING $S$ IN PRACTICE

Given a training set $\mathcal{X}$ of in-distribution samples and the corresponding scores $S(\mathbf{x})$ for each $\mathbf{x} \in \mathcal{X}$, we foresee a number of strategies to perform OOD detection in practice for new instances $\mathbf{z}$. The first and more straightforward one is to use $S(\mathbf{z})$ as it is, just as a score, to perform OOD ranking. This can be useful to monitor the top-k, potentially more problematic instances $\mathbf{z}$ in a new set of unlabeled data $\mathcal{Z}$. The second strategy is to interpret $S(\mathbf{z})$ as the corresponding Bayes factor in Eq. 3, and directly assign $\mathbf{z}$ to be OOD for $S(\mathbf{z}) > 0$, or in-distribution otherwise (cf. MacKay, 2003). The decision is then taken with stronger evidence for higher absolute values of $S(\mathbf{z})$. A third strategy is to consider the empirical or null distribution of $S$ for the full training set, $S(\mathcal{X})$. We could then choose an appropriate quantile as threshold, adopting the notion of frequentist hypothesis testing (see for instance Nalisnick et al., 2019b). Finally, if ground truth OOD data $\mathcal{Y}$ is available, a fourth strategy is to optimize a threshold value for $S(\mathbf{z})$. Using $\mathcal{X}$ and $\mathcal{Y}$, we can choose a threshold that targets a desired percentage of false positives or negatives.

The choice of a specific strategy will depend on the characteristics of the particular application under consideration. In this work, we prefer to keep our evaluation generic and to not adopt any specific thresholding strategy (that is, we use $S$ directly, as a score). This also allows us to compare with the majority of reported values from the existing literature, which use the AUROC measure (see below).

## 4 RELATED WORKS

Ren et al. (2019) have recently proposed the use of likelihood-ratio tests for OOD detection. They posit that "background statistics" (for instance, the number of zeros in the background of MNIST-like images) are the source of abnormal likelihoods, and propose to exploit them by learning a background model which is trained on random surrogates of input data. Such surrogates are generated according to a Bernoulli distribution, and an L2 regularization term is added to the background

model, which implies that the approach has two hyper-parameters. Moreover, both the background model and the model trained using in-distribution data need to capture the background information equally well. In contrast to their method, our method does not require additional training nor extra conditions on a specific background model for every type of training data.

Choi et al. (2018) and Nalisnick et al. (2019b) suggest that typicality is the culprit for likelihood-based generative models not being able to detect OOD inputs. While Choi et al. (2018) do not explicitly address typicality, their estimate of the Watanabe-Akaike information criterion using ensembles of generative models performs well in practice. Nalisnick et al. (2019b) propose an explicit test for typicality employing a Monte Carlo estimate of the empirical entropy, which limits their approach to batches of inputs of the same type.

The works of Høst-Madsen et al. (2019) and Sabeti & Høst-Madsen (2019) combine the concepts of typicality and minimum description length to perform novelty detection. Although concepts are similar to the ones employed here, their focus is mainly on bit sequences. They consider atypical sequences those that can be described (coded) with fewer bits in itself rather than using the (optimum) code for typical sequences. We find their implementation to rely on strong parametric assumptions, which makes it difficult to generalize to generative or other machine learning models.

A number of methods have been proposed to perform OOD detection under a classification-based framework (Hendrycks & Gimpel, 2017; Lakshminarayanan et al., 2017; Liang et al., 2018; Alemi et al., 2018; Lee et al., 2018; Hendrycks et al., 2019). Although achieving promising results, these methods do not generally apply to the more general case of non-labeled or self-supervised data. The method of Hendrycks et al. (2019) extends to such cases by leveraging generative models, but nonetheless makes use of auxiliary, outlier data to learn to distinguish OOD inputs.

## 5  RESULTS

We now study how $S$ performs on the OOD detection task. For that, we train a generative model $\mathcal{M}$ on the train partition of a given data set and compute scores for such partition and the test partition of a different data set. With both sets of scores, we then calculate the area under the receiver operating characteristic curve (AUROC), which is a common evaluation measure for the OOD detection task (Hendrycks et al., 2019) and for classification tasks in general. Note that AUROC represents a good performance summary across different score thresholds (Fawcett, 2005).

First of all, we want to assess the improvement of $S$ over log-likelihoods alone ($-\ell_{\mathcal{M}}$). When considering likelihoods from generative models trained on CIFAR10, the problematic results reported by previous works become clearly apparent (Table 1). The unintuitive higher likelihoods for SVHN observed in Sec. 1 now translate into a poor AUROC below 0.1. This not only happens for SVHN, but also for Constant, Omniglot, MNIST, and FashionMNIST data sets, for which we observed consistently higher likelihoods than CIFAR10 in Sec. 2. Likelihoods for the other data sets yield AUROCs above the random baseline of 0.5, but none above 0.67. The only exception is the Noise data set, which is perfectly distinguishable from CIFAR10 using likelihood alone. For completeness, we include the AUROC values when trying to perform OOD with the test partition of CIFAR10. We see those are close to the ideal value of 0.5, showing that, as expected, the reported measures do not generally consider those samples to be OOD.

We now look at the AUROCs obtained with $S$ (Table 1). We see that, not only results are reversed for less complex datasets like MNIST or SVHN, but also that all AUROCs for the rest of the data sets improve as well. The only exception to the last assertion among all studied combinations is the combination of TinyImageNet with PixelCNN++ and FLIF (see Appendix D for other combinations). In general, we obtain AUROCs above 0.7, with many of them approaching 0.9 or 1. Thus, we can conclude that $S$ clearly improves over likelihoods alone in the OOD detection task, and that $S$ is able to revert the situation with intuitively less complex data sets that were previously yielding a low AUROC.

We also study how the training set, the choice of compressor/generative model, or the size of the model affects the performance of $S$ (Appendix D). In terms of models and compressors, we do not observe a large difference between the considered combinations, except for a few isolated cases whose investigation we defer for future work. In terms of model size, we do observe a tendency to provide better discrimination with increasing size. In terms of data sets, we find the OOD detection

Table 1: AUROC values using negative log-likelihood $-\ell_{\mathcal{M}}$ and the proposed score $S$ for Glow and PixelCNN++ models trained on CIFAR10 using the FLIF compressor. Results for models trained on FashionMNIST and other compressors are available in Appendix D.

| Data set | Glow | | PixelCNN++ | |
|---|---|---|---|---|
| | $-\ell_{\mathcal{M}}$ | $S$ | $-\ell_{\mathcal{M}}$ | $S$ |
| Constant | 0.024 | 1.000 | 0.006 | 1.000 |
| Omniglot | 0.001 | 1.000 | 0.001 | 1.000 |
| MNIST | 0.001 | 1.000 | 0.002 | 1.000 |
| FashionMNIST | 0.010 | 1.000 | 0.013 | 1.000 |
| SVHN | 0.083 | 0.950 | 0.083 | 0.929 |
| CIFAR100 | 0.582 | 0.736 | 0.526 | 0.535 |
| CelebA | 0.621 | 0.863 | 0.624 | 0.776 |
| FaceScrub | 0.646 | 0.859 | 0.643 | 0.760 |
| TinyImageNet | 0.663 | 0.716 | 0.642 | 0.589 |
| TrafficSign | 0.609 | 0.931 | 0.599 | 0.870 |
| Noise | 1.000 | 1.000 | 1.000 | 1.000 |
| *CIFAR10 (test)* | *0.564* | *0.618* | *0.506* | *0.514* |

task to be easier with FashionMNIST than with CIFAR10. We assume that this is due to the ease of the generative model to learn and approximate the density conveyed by the data. A similar but less marked trend is also observed for compressors, with better compressors yielding slightly improved AUROCs than other, in principle, less powerful ones. A takeaway from all these observations would be that using larger generative models and better compressors will yield a more reliable $S$ and a better AUROC. The conducted experiments support that, but a more in-depth analysis should be carried out to further confirm this hypothesis.

Finally, we want to assess how $S$ compares to previous approaches in the literature. For that, we compile a number of reported AUROCs for both classifier- and generative-based approaches and compare them with $S$. Note that classifier-based approaches, as mentioned in Sec. 1, are less applicable than generative-based ones. In addition, as they exploit label information, they might have an advantage over generative-based approaches in terms of performance (some also exploit external or outlier data; Sec. 4).

We observe that $S$ is competitive with both classifier- and existing generative-based approaches (Table 2). When training with FashionMNIST, $S$ achieves the best scores among all considered approaches. The results with further test sets are also encouraging, with almost all AUROCs approaching 1 (Appendix D). When training with CIFAR10, $S$ achieves similar or better performance than existing approaches. Noticeably, within generative-based approaches, $S$ is only outperformed in two occasions by the same approach, WAIC, which uses ensembles of generative models (Sec. 4).

On the one hand, it would be interesting to see how $S$ could perform when using ensembles of models and compressors to produce better estimates of $-\ell_{\mathcal{M}}$ and $L$, respectively. On the other hand, however, the use of a single generative model together with a single fast compression library makes $S$ an efficient alternative compared to WAIC and some other existing approaches. It is also worth noting that many existing approaches have a number of hyper-parameters that need to be tuned, sometimes with the help of outlier or additional data. In contrast, $S$ is a parameter-free measure, which makes it easy to use and deploy.

## 6 CONCLUSION

We illustrate a fundamental insight with regard to the use of generative models' likelihoods for the task of detecting OOD data. We show that input complexity has a strong effect in those likelihoods, and pose that it is the main culprit for the puzzling results of using generative models' likelihoods for OOD detection. In addition, we show that an estimate of input complexity can be used to com-

Table 2: Comparison of AUROC values for the OOD detection task. Results as reported by the original references except (a) by Ren et al. (2019), (b) by Lee et al. (2018), and (c) by Choi et al. (2018). Results for Typicality test correspond to using batches of 2 samples of the same type.

| Trained on: | FashionMNIST | | CIFAR10 | | |
|---|---|---|---|---|---|
| OOD data: | MNIST | Omniglot | SVHN | CelebA | CIFAR100 |
| *Classifier-based approaches* | | | | | |
| ODIN (Liang et al., 2018)[a,b] | 0.697 | - | 0.966 | - | - |
| VIB (Alemi et al., 2018)[c] | 0.941 | 0.943 | 0.528 | 0.735 | - |
| Mahalanobis (Lee et al., 2018) | 0.986 | - | 0.991 | - | - |
| Outlier exposure (Hendrycks et al., 2019) | - | - | 0.984 | - | **0.933** |
| *Generative-based approaches* | | | | | |
| WAIC (Choi et al., 2018) | 0.766 | 0.796 | **1.000** | **0.997** | - |
| Outlier exposure (Hendrycks et al., 2019) | - | - | 0.758 | - | 0.685 |
| Typicality test (Nalisnick et al., 2019b) | 0.140 | - | 0.420 | - | - |
| Likelihood-ratio (Ren et al., 2019) | 0.997 | - | 0.912 | - | - |
| $S$ using Glow and FLIF (ours) | **0.998** | **1.000** | 0.950 | 0.863 | 0.736 |
| $S$ using PixelCNN++ and FLIF (ours) | 0.967 | 1.000 | 0.929 | 0.776 | 0.535 |

pensate standard negative log-likelihoods in order to produce an efficient and reliable OOD score. We also offer an interpretation of our score as a likelihood-ratio akin to Bayesian model comparison. Such score performs comparably to, or even better than several state-of-the-art approaches, with results that are consistent across a range of data sets, models, model sizes, and compression algorithms. The proposed score has no hyper-parameters besides the definition of a generative model and a compression algorithm, which makes it easy to employ in a variety of practical problems and situations.

ACKNOWLEDGMENTS

We thank Ilias Leontiadis and Santi Pascual for useful discussions at the beginning of this project. Vicenç Gómez is supported by the Ramon y Cajal program RYC-2015-18878 (AEI/MINEICO/FSE,UE). This project has been partially funded by "la Caixa" Foundation (ID 100010434) under agreement LCF/PR/PR16/51110009 and by the EU's Horizon 2020 research and innovation program under grant agreement No 871793. This text reflects only the authors' view and the Commission is not responsible for any use that may be made of the information it contains.

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

APPENDIX

## A  DATA SETS

In our experiments, we employ well-known, publicly-available data sets. In addition to those, and to facilitate a better understanding of the problem, we develop another two self-created sets of synthetic images: Noise and Constant images. The Noise data set is created by uniformly randomly sampling a tensor of $3\times32\times32$ and quantizing the result to 8 bits. The Constant data set is created similarly, but using a tensor of $3\times1\times1$ and repeating the values along the last two dimensions to obtain a size of $3\times32\times32$. The complete list of data sets is available in Table 3. In the case of data sets with different variations, such as CelebA or FaceScrub, which have both plain and aligned versions of the faces, we select the aligned versions. Note that, for models trained on CIFAR10, it is important to notice the overlap of certain classes between that and other sets, namely TinyImageNet and CIFAR100 (they overlap, for instance, in classes of certain animals or vehicles). Therefore, strictly speaking, such data sets are not entirely OOD, at least semantically.

Table 3: Summary of the considered data sets.

| Data set | Original size | Num. classes | Num. images |
|---|---|---|---|
| Constant (Synthetic) | $3\times32\times32$ | 1 | 40,000 |
| Omniglot (Lake et al., 2015) | $1\times105\times105$ | 1,623 | 32,460 |
| MNIST (LeCun et al., 2010) | $1\times28\times28$ | 10 | 70,000 |
| FashionMNIST (Xiao et al., 2017) | $1\times28\times28$ | 10 | 70,000 |
| SVHN (Netzer et al., 2011) | $3\times 32\times32$ | 10 | 99,289 |
| CIFAR10 (Krizhevsky, 2009) | $3\times32\times32$ | 10 | 60,000 |
| CIFAR100 (Krizhevsky, 2009) | $3\times32\times32$ | 100 | 60,000 |
| CelebA (Liu et al., 2015) | $3\times178\times218$ | 10,177 | 182,732 |
| FaceScrub (Ng & Winkler, 2014) | $3\times300\times300$ | 530 | 91,712 |
| TinyImageNet (Deng et al., 2009) | $3\times64\times64$ | 200 | 100,000 |
| TrafficSign (Stallkamp et al., 2011) | $3\times32\times32$ | 43 | 51,839 |
| Noise (Synthetic) | $3\times32\times32$ | 1 | 40,000 |

In order to split the data between train, validation, and test, we follow two simple rules: (1) if the data set contains some predefined train and test splits, we respect them and create a validation split using a random 10% of the training data; (2) if no predefined splits are available, we create them by randomly assigning 80% of the data to the train split and 10% to both validation and test splits. In order to create consistent input sizes for the generative models, we work with 3-channel images of size $32\times32$. For those data sets which do not match this configuration, we follow a classic bi-linear resizing strategy and, to simulate the three color components from a gray-scale image, we triplicate the channel dimension.

## B  MODELS AND TRAINING

The results of this paper are obtained using two generative models of different nature: one autoregressive model and one invertible model. As autoregressive model we choose PixelCNN++ (Salimans et al., 2017), which has been shown to obtain very good results in terms of likelihood for image data. As invertible model we choose Glow (Kingma & Dhariwal, 2018), which is also capable of inferring exact log-likelihoods using large stacks of bijective transformations. We implement the Glow model using the default configuration of the original implementation[1], except that we zero-pad and do not use ActNorm inside the coupling network. The model has 3 blocks of 32 flows, using an affine coupling with an squeezing factor of 2. As for PixelCNN++, we set 5 residual blocks per

---

[1] https://github.com/openai/glow

stage, with 80 filters and 10 logistic components in the mixture. The non-linearity of the residual layers corresponds to an exponential linear unit[2].

We train both Glow and PixelCNN++ using the Adam optimizer with an initial learning rate of $10^{-4}$. We reduce this initial value by a factor of $1/5$ every time that the validation loss does not decrease during 5 consecutive epochs. The training finishes when the learning rate is reduced by factor of $1/100$. The batch size of both models is set to 50. The final model weights are the ones yielding the best validation loss. The likelihoods obtained in validation with both Glow and PixelCNN++ match the ones reported in the literature for CIFAR10 (Kingma & Dhariwal, 2018; Salimans et al., 2017). We also make sure that the generated images are of comparable quality to the ones shown in those references.

We use PyTorch version 1.2.0 (Paszke et al., 2017). All models have been trained with a single NVIDIA GeForce GTX 1080Ti GPU. Training takes some hours under that setting.

## C    COMPRESSORS AND COMPLEXITY ESTIMATE

We explore three different options to compress input images. As a mandatory condition, they need to provide lossless compression. The first format that we consider is PNG, and old-classic format which is globally used and well-known. We use OpenCV[3] to compress from raw Numpy matrices, with compression set to the maximum possible level. The second format that we consider is JPEG2000. Although not as globally known as the previous one, it is a more modern format with several new generation features such as progressive decoding. Again, we use the default OpenCV implementation to obtain the size of an image using this compression algorithm. The third format that we consider is FLIF, the most modern algorithm of the list. According to its website[4], it promises to generate up to 53% smaller files than JPEG2000. We use the publicly-available compressor implementation in their website. We do not include header sizes in the measurement of the resulting bits per dimension.

To compute our complexity estimate $L(\mathbf{x})$, we compress the input $\mathbf{x}$ with one of the compressors $C$ above. With that, we obtain a string of bits $C(\mathbf{x})$. The length of it, $|C(\mathbf{x})|$, is normalized by the size or dimensionality of $\mathbf{x}$, which we denote by $d$, to obtain the complexity estimate:

$$L(\mathbf{x}) = \frac{|C(\mathbf{x})|}{d}.$$

We also experimented with an improved version of $L$,

$$L'(\mathbf{x}) = \min\left(L_1(\mathbf{x}), L_2(\mathbf{x}), \dots\right),$$

where $L_i$ corresponds to different compression schemes. This forces $S$ to work always with the best compressor for every $\mathbf{x}$. In our case, as FLIF was almost always the best compressor, we did not observe a clear difference between using $L$ or $L'$. However, in cases where it is not clear which compressor to use or cases in which we do not have a clear best/winner, $L'$ could be of use.

## D    ADDITIONAL RESULTS

The additional results mentioned in the main paper are the following:

- In Table 4, we report the average log-likelihood $\bar{\ell}_{\mathcal{M}}$ for every data set. We sort data sets from highest to lowest log-likelihood.

- In Table 5, we report the global Pearson's correlation coefficient for different models, train sets, and compressors. Due to the large sample size, Scipy version 1.2.1 reports a p-value of 0 in all cases.

- In Table 6, we report the AUROC values obtained from log-likelihoods $\ell_{\mathcal{M}}$, complexity estimates $L$, a simple two-tail test $T$ taking into account lower and higher log-likelihoods, $T = |\bar{\ell}_{\mathcal{M}} - \ell_{\mathcal{M}}|$, and the proposed score $S$.

---

[2] https://github.com/pclucas14/pixel-cnn-pp
[3] https://opencv.org
[4] https://flif.info

- In Table 7, we report the AUROC values obtained from $S$ across different Glow model sizes, using a PNG compressor.
- In Table 8, we report the AUROC values obtained from $S$ across different data sets, models, and compressors.

Table 4: Average log-likelihoods from a PixelCNN++ model trained on CIFAR10.

| Data set | $\overline{\ell}_{\mathcal{M}}$ |
|---|---|
| Constant (Test) | $-0.25$ |
| Omniglot (Test) | $-0.43$ |
| MNIST (Test) | $-0.55$ |
| FashionMNIST (Test) | $-0.83$ |
| SVHN (Test) | $-1.19$ |
| CIFAR10 (Train) | $-2.20$ |
| CIFAR10 (Test) | $-2.21$ |
| CIFAR100 (Test) | $-2.27$ |
| CelebA (Test) | $-2.42$ |
| FaceScrub (Test) | $-2.43$ |
| TinyImageNet (Test) | $-2.51$ |
| TrafficSign (Test) | $-2.51$ |
| Noise (Test) | $-8.22$ |

Table 5: Pearson's correlation coefficient between normalized compressed length, using different compressors, and model likelihood. All correlations are statistically significant (see text).

| Model | Trained with | Compressor | | |
|---|---|---|---|---|
| | | PNG | JPEG2000 | FLIF |
| Glow | FashionMNIST | $-0.77$ | $-0.75$ | $-0.77$ |
| PixelCNN++ | FashionMNIST | $-0.77$ | $-0.77$ | $-0.78$ |
| Glow | CIFAR10 | $-0.94$ | $-0.90$ | $-0.90$ |
| PixelCNN++ | CIFAR10 | $-0.96$ | $-0.94$ | $-0.94$ |

Table 6: AUROC values using negative log-likelihood $-\ell_{\mathcal{M}}$, the complexity measure $L$, a simple two-tail test $T$ (see text), and our score $S$ for Glow and PixelCNN++ models trained on CIFAR10 and using a PNG compressor. Qualitatively similar results were obtained for FashionMNIST and other compressors.

| Data set | Glow | | | | PixelCNN++ | | | |
|---|---|---|---|---|---|---|---|---|
| | $-\ell_{\mathcal{M}}$ | $L$ | $T$ | $S$ | $-\ell_{\mathcal{M}}$ | $L$ | $T$ | $S$ |
| Constant | 0.024 | 0.000 | 0.963 | 1.000 | 0.006 | 0.000 | 0.987 | 1.000 |
| Omniglot | 0.001 | 0.000 | 0.999 | 1.000 | 0.001 | 0.000 | 0.995 | 1.000 |
| MNIST | 0.001 | 0.000 | 0.998 | 1.000 | 0.002 | 0.000 | 0.992 | 1.000 |
| FashionMNIST | 0.010 | 0.003 | 0.987 | 1.000 | 0.013 | 0.003 | 0.966 | 1.000 |
| SVHN | 0.083 | 0.077 | 0.845 | 0.950 | 0.083 | 0.077 | 0.832 | 0.929 |
| CIFAR100 | 0.582 | 0.483 | 0.576 | 0.736 | 0.526 | 0.483 | 0.540 | 0.535 |
| CelebA | 0.621 | 0.414 | 0.458 | 0.863 | 0.624 | 0.414 | 0.414 | 0.776 |
| FaceScrub | 0.646 | 0.452 | 0.472 | 0.859 | 0.643 | 0.452 | 0.425 | 0.760 |
| TinyImageNet | 0.663 | 0.548 | 0.585 | 0.716 | 0.642 | 0.548 | 0.544 | 0.589 |
| TrafficSign | 0.609 | 0.356 | 0.689 | 0.931 | 0.599 | 0.357 | 0.657 | 0.870 |
| Noise | 1.000 | 1.000 | 1.000 | 1.000 | 1.000 | 1.000 | 1.000 | 1.000 |

Table 7: AUROC values for $S$ using a Glow model trained on CIFAR10 and a PNG compressor. Results for different, increasing sizes of the model (blocks $\times$ flow steps). Qualitatively similar results are obtained for other compressors.

| Data set | $2\times16$ | $3\times16$ | $3\times32$ |
|---|---|---|---|
| Constant | 1.000 | 1.000 | 1.000 |
| Omniglot | 1.000 | 1.000 | 1.000 |
| MNIST | 1.000 | 1.000 | 1.000 |
| FashionMNIST | 0.997 | 0.998 | 1.000 |
| SVHN | 0.765 | 0.783 | 0.950 |
| CIFAR100 | 0.641 | 0.685 | 0.736 |
| CelebA | 0.741 | 0.794 | 0.863 |
| FaceScrub | 0.697 | 0.755 | 0.859 |
| TinyImageNet | 0.664 | 0.715 | 0.716 |
| TrafficSign | 0.946 | 0.957 | 0.931 |
| Noise | 1.000 | 1.000 | 1.000 |

Table 8: Comparison of AUROC values for the proposed OOD score $S$ using different compressors: Glow and PixelCNN++ models trained on FashionMNIST (top) and CIFAR10 (bottom).

| Data set | Glow | | | PixelCNN++ | | |
|---|---|---|---|---|---|---|
| | PNG | JPEG2000 | FLIF | PNG | JPEG2000 | FLIF |
| Constant | 1.000 | 1.000 | 1.000 | 1.000 | 1.000 | 1.000 |
| Omniglot | 1.000 | 1.000 | 1.000 | 1.000 | 1.000 | 1.000 |
| MNIST | 0.841 | 0.493 | 0.997 | 0.821 | 0.687 | 0.967 |
| SVHN | 1.000 | 1.000 | 1.000 | 1.000 | 1.000 | 1.000 |
| CIFAR10 | 1.000 | 1.000 | 1.000 | 0.998 | 1.000 | 1.000 |
| CIFAR100 | 1.000 | 1.000 | 1.000 | 0.997 | 1.000 | 1.000 |
| CelebA | 1.000 | 1.000 | 1.000 | 1.000 | 1.000 | 1.000 |
| FaceScrub | 1.000 | 1.000 | 1.000 | 1.000 | 1.000 | 1.000 |
| TinyImageNet | 1.000 | 1.000 | 1.000 | 1.000 | 1.000 | 1.000 |
| TrafficSign | 1.000 | 1.000 | 1.000 | 1.000 | 1.000 | 1.000 |
| Noise | 1.000 | 1.000 | 1.000 | 1.000 | 1.000 | 1.000 |
| Constant | 1.000 | 1.000 | 1.000 | 1.000 | 1.000 | 1.000 |
| Omniglot | 1.000 | 0.994 | 1.000 | 1.000 | 0.997 | 1.000 |
| MNIST | 1.000 | 0.996 | 1.000 | 1.000 | 0.995 | 1.000 |
| FashionMNIST | 0.998 | 0.998 | 1.000 | 0.998 | 0.995 | 1.000 |
| SVHN | 0.787 | 0.974 | 0.950 | 0.787 | 0.965 | 0.929 |
| CIFAR100 | 0.683 | 0.757 | 0.736 | 0.583 | 0.514 | 0.535 |
| CelebA | 0.794 | 0.701 | 0.863 | 0.756 | 0.640 | 0.776 |
| FaceScrub | 0.750 | 0.797 | 0.859 | 0.710 | 0.704 | 0.760 |
| TinyImageNet | 0.710 | 0.875 | 0.716 | 0.657 | 0.735 | 0.589 |
| TrafficSign | 0.953 | 0.955 | 0.931 | 0.916 | 0.840 | 0.870 |
| Noise | 1.000 | 1.000 | 1.000 | 1.000 | 1.000 | 1.000 |

