# OpenReview forum: "Input Complexity and Out-of-distribution Detection with Likelihood-based Generative Models"
_ICLR.cc/2020/Conference — Accept (Poster)_

### Official Review · AnonReviewer1 · 2019-10-21
**Official Blind Review #1**

**Rating:** 3

**Review:**

The paper discusses the inductive biases in generative models that tend to assign higher likelihoods to "less complex" images. In particular, a likelihood based generative model (like Glow or PixelCNN++) trained on a particular dataset has significantly higher likelihood for data that have lower compression ratios (e.g  with PNG). The authors propose a simple approach based on the likelihood ratio between a trained model and a "prior" model (based on existing compression methods) and demonstrate that this improves unsupervised OOD detection on certain dataset pairs. The idea is quite simple (and surprising it seems to work!), but it seems that better understanding of the proposed method could be achieved.

Questions:

If our goal is to perform OOD detection, then higher likelihood for test samples might not be an issue, as we can simply declare samples with higher and lower likelihoods as OOD?

It would seem that using L(x) might already distinguish some OOD samples from Figure 4? What are the AUC of OOD detection if we had simply used these?

From Figure 4, the x axis is p(x|M) and is between zero and one, while L(x) should be -log p(x|M_0). Why would we expect to observe a highly linear correlation between the two (as opposed to linear with p(x|M) and exp(L(x)))? Visually it seems that a lot of the datasets have similar distributions in both cases -- wonder why the proposed S statistic improves OOD empirically? Does this still hold for continuous likelihood models like Glow?

The performance seems to depend on the "prior" chosen (e.g. FLIF seems to have much better performance than PNG); any insights why this is the case?

Table 1: I suppose a reasonable comparison for AUROC is max(current, 1 - current)? If AUROC is very small, we can still obtain good classifiers with flipped predictions.

It seems strange that the black-and-white images and colored images are compared together (say I train on MNIST and test CIFAR10); it should be quite easy to detect even with existing approaches. I suppose it would be most interesting to see cases where existing approaches fail miserably (like CIFAR10 vs. CIFAR100/TinyImageNet); unfortunately the proposed methods does not seem to improve much for PixelCNN++.


**Experience Assessment:**

I have read many papers in this area.

**Review Assessment: Checking Correctness Of Derivations And Theory:**

I carefully checked the derivations and theory.

**Review Assessment: Checking Correctness Of Experiments:**

I carefully checked the experiments.

**Review Assessment: Thoroughness In Paper Reading:**

I read the paper at least twice and used my best judgement in assessing the paper.

---

> ### Author Response · Authors · 2019-11-08
> **Response to Review #1**
>
> We thank Reviewer 1 for his/her detailed feedback. In the following, we answer or clarify all questions raised by the Reviewer (Q1-Q6).
>
> Q1: Using higher and lower likelihoods — We believe this solution is not general enough, as it would fail in cases with overlapping distributions of likelihoods. Also, contrary to our method, it does not provide any intuitive argument about the behavior of generative models’ likelihoods with OOD data. Regarding the first point, we note that there is always a target distribution that completely overlaps with the training distribution (see figures 1,2 and references [1,2]). Nevertheless, we have done a preliminary analysis and found that, while outperforming the naive likelihood-based method, this solution performs worse than our proposed method. We thank the reviewer for the insight, and have now added these results in Appendix D. Finally, notice that reference [2] also considers related solutions, such as typical two-tail tests on the empirical likelihoods, that perform worse than our method.
>
> Q2: Use of L(X) — We empirically found that using L(x) works as bad as the original likelihood p(x|M). This is expected, given the high correlation reported in Table 5 (Appendix D). We thank this suggestion and, for completeness, we have now added these results in the revised version (Table 6 in Appendix D).
>
> Q3: Correlation between likelihood and L(x) or exp(L(x)) — Figure 4 has the only purpose of illustrating the identified bias, showing the strong correlation between the complexity and the likelihoods. We postulate that accounting for such a bias is critical for OOD. Whether we use an exp(L(x)) or L(x) is a minor detail, as p(x|M) and p(x|M0) also show a strong correlation. The same holds for Glow. We hope this is clarified now.
>
> Q4: Insights on compressor — As mentioned in the paper (and shown in Appendix D), better compressors typically result in more accurate estimates of L(x), which should therefore facilitate the discrimination of OOD inputs. In many cases, however, the differences are minor, probably because of similar compression ratios for different compressors. Thus, in those cases, we could say that the choice of the compressor is non-critical (as long as it is lossless).
>
> Q5: AUROC flipped predictions — The problem here is that, at test time (and potentially with a single sample), there is no way to decide whether we should flip the decision of the detector or not. If the behavior was consistent across all data sets (say, all data sets were yielding AUCs close to 0), then we could just assume the classifier decision is always flipped. However, in the considered problem, the behavior is not consistent across data sets nor samples. Since, at test time, we do not know the dataset to which a specific sample corresponds to, we cannot perform such flipping.
>
> Q6: The difficulty of distinguishing grayscale from color images with likelihood-based generative models is well documented in the literature [1,2]. In addition, results for existing approaches are not typically reported for CIFAR10 vs CIFAR100/TinyImageNet. That is the main reason why we do not include those in Table 2. In a sense, we could agree that the latter are challenging cases but, at the same time, one should notice that CIFAR100/TinyImageNet have a number of overlapping categories with CIFAR10 (for instance, animals’ or vehicles’ categories). Thus, perhaps we should caution about using such specific benchmark for OOD, as some samples of CIFAR100/TinyImageNet may not be, strictly speaking, OOD instances with respect to CIFAR10.
>
> [1] H. Choi, E. Jang, and A. A. Alemi. “WAIC, but why? Generative ensembles for robust anomaly detection”. ArXiv, 1810.01392, 2018.
> [2] E. Nalisnick, A. Matsukawa, Y. W. Teh, and B. Lakshminarayanan. “Detecting out-of-distribution inputs to deep generative models using a test for typicality”. ArXiv, 1906.02994, 2019.

---

### Official Review · AnonReviewer2 · 2019-10-23
**Official Blind Review #2**

**Rating:** 3

**Review:**

# ==== Summary of the paper ====

This paper proposes a new criterion for out-of-distribution (OOD) detection for
generative modelling of images (absence of labels). The OOD detection task is defined as follows: given a generative model p(x) trained on some data $X \sim p^*$ (where $p^*$ is the true unknown data generating distribution), and a test point y, how can we make use of p to detect that y is not drawn from $p^*$ (i.e., out of distribution)? The first natural idea is to use the likelihood (density) given by p, which, by now, is known to be problematic since the likelihood can be high when evaluated on data points from a completely different domain.

This paper contributes the following results:

1. Show that "simple" images (e.g., constant color) tend to give high likelihood, whereas complex images (e.g., noise) tend to give low likelihood. See Figure 2. The paper further quantifies this complexity with the normalized size of the compressed input images, as given by a compression algorithm. The paper shows that there is a negative correlation between the compressed size and the likelihood (models trained on CIFAR 10. See Figure 4).

2. To address this issue with using only the likelihood for OOD detection, the paper proposes a new measure $S(x)$ given by the negative log likelihood minus the normalized compressed size. See Eq 1. The paper connects this measure to Bayesian model selection.

Empirical results on more than 10 image datasets show that the proposed measure works better than the negative log likelihood in most cases.

# ==== Review ====

The paper is easy to follow. The finding that simple images tend to give high likelihood is interesting. My concerns are:

1. How much does the conclusion that "simple images tend to give high likelihood" depend on the complexity of the model? As far as I can see, only a few models are studied here: Glow and PixelCNN++. What is the quality of the learned models? Do they generate realistic images? What happens if you consider simpler models (say, reduce the number of layers in Glow.)?

2. Given a model p and a test input image y, how exactly do you tell if y is out of distribution? Is there a threshold? If so, what is the threshold? I understand that by AUROC used in Table 1, you vary the threshold. For each value of the threshold, you compute the true positive and false positive rates, and plot the ROC curve. The reported numbers in Table 1 are areas under the curve. Is this correct? But this does not explain how to perform OOD detection given one input image.

3. What is the reason for the poor AUROC in the case of TinyImageNet in Table 1? This is an interesting case since it may suggest that there are other hidden factors (besides the complexity of images) that can affect the OOD detection with the likelihood.

4. Have you tried to run the experiment in Table 1 with a model trained on Constant (simplest images) or Noise (most complex images)?  What happens? Also, why not also include CIFAR10 (used to train the model) in the list of datasets in Table 1?

Overall, my main concerns are with the thoroughness of the experiments, and that the two main contributions (summarized above) may not be enough. I will consider my evaluation again after seeing responses from the authors.


# ==== Minor. Did not affect the score ====

* The bottom margin seems off. Please check whether the Latex template is used correctly.

* The paragraph before Section 3: "... could replace the computed likelihood values for the negative of our complexity estimate ..." I think it is too soon to make this conclusion.

* The term "likelihood ratio test" used in Section 3.2 is very misleading. There is no hypothesis testing there.

* Might be better to briefly describe the meaning of AUROC at the beginning of section 5.






**Experience Assessment:**

I have read many papers in this area.

**Review Assessment: Checking Correctness Of Derivations And Theory:**

N/A

**Review Assessment: Checking Correctness Of Experiments:**

I assessed the sensibility of the experiments.

**Review Assessment: Thoroughness In Paper Reading:**

I read the paper at least twice and used my best judgement in assessing the paper.

---

> ### Author Response · Authors · 2019-11-08
> **Response to Review #2**
>
> We thank Reviewer 2 for his/her insightful comments. We now answer and discuss the 4 main concerns expressed by the Reviewer (C1-C4). We appreciate the minor suggestions from the Reviewer and have considered them in the revised version.
>
> C1: Model complexity and quality of learned models — Both Glow and PixelCNN++ models yield validation/test likelihoods that are comparable to the ones reported in the original references on CIFAR10. We also did an informal check to be sure that generated images resembled the ones reported in those references. We have added a couple of sentences stating this in Appendix A. Regarding the relation between model complexity and OOD detection accuracy, we have run additional experiments with the Glow model as suggested, removing blocks and flow steps, and observe a trend towards better accuracy with larger/more capable models. This additionally supports our original hypothesis that larger/more capable models and compressors provide a better measure S(x). We have incorporated the new results in Appendix D and commented on them in the main paper. We deeply thank the Reviewer for the suggestion.
>
> C2: OOD evaluation and AUROC — There are two main ways to perform OOD detection from S(x) in practice. The first one is to compute S(x) for all training data, and then select a threshold based on the empirical distribution of S(x), taking into account the percentage of false negatives that we can tolerate for a particular application. The second one is to compute S(x) for all training data and some confirmed OOD data, and then select a threshold based on the percentage of false positives and false negatives required by a particular application. We now clarify in the text that S(x) corresponds to a test statistic and apologize for any misunderstanding this might have caused. As correctly stated by the Reviewer, the reported numbers correspond to the area under the ROC curve (AUROC). AUROC allows to evaluate the performance of a system for a general case, without the need of establishing a threshold value. In the case where we are not sure whether false positives or false negatives are better or worse for a particular application, AUROC provides a useful single-number summary of the performance of a classifier [1]. AUROC is also the most common evaluation measure in OOD detection (see for instance [2]) and, as such, is one of the few means to compare across the performances reported by the state-of-the-art.
>
> C3: TinyImageNet data — The poorest AUROCs in Table 1 are obtained for TinyImageNet and CIFAR100. We explain those by the fact that both data sets have a number of classes or categories that conceptually overlap with the ones of CIFAR10 (for instance, animals’ or vehicles’ categories). Therefore, strictly speaking, those data sets are not entirely OOD with respect to CIFAR10. We mention this issue only in Appendix A. However, we have no objection to include this in the main paper for another revised version.
>
> C4: Constant, Noise, and CIFAR10 results — We have the impression that experiments related to models trained with Constant and Noise images may be unrealistic and not very informative. Nonetheless, we trained Glow models with Constant and Noise images and ran experiments as the one in Table 1. For Noise images, likelihoods yield an average AUROC of 0.936 across data sets, and S(x) yields an AUROC of 1.000 in all cases. Therefore, although likelihoods alone achieve a good performance, S(x) is still able to improve it. For Constant images, all AUROCs are equal to 1 for all data sets for both likelihoods and S(x). Therefore, as there is already no improvement possible, we do not observe any difference. Regarding CIFAR10 results, we now include them in Table 1, and thank the Reviewer for the suggestion. We didn’t do it in the first place because we thought they may cause some confusion to the readers, as they are from an in-sample (not OOD) data set.
>
> [1] T. Fawcett. An introduction to ROC analysis. Pattern Recognition Letters 27(8): 861-874, 2006.
> [2] D. Hendrycks, M. Mazeika, and T.G. Dietterich. “Deep anomaly detection with outlier exposure”. Proc. of the Int. Conf. on Learning Representations (ICLR), 2019.

---

> > ### Comment · AnonReviewer2 · 2019-11-14
> > **Thanks for your responses**
> >
> > Thank you for your responses and additional experiments. I think my biggest concern, C2, has not been addressed. Are you suggesting that, to determine the threshold, firstly one has to have some confirmed OOD examples so we can pick a well-calibrated threshold? Then, it sounds like you are solving a kind of a chicken-and-egg problem? Deciding whether some examples are OOD is your original task. Now, to do so, first, you need to have some OOD examples. I am not sure about this idea in practice. Of courses, you can prepare a set of images from a completely different dataset and treat those as OOD. But in practice, I believe the interesting case is when you have images that are borderline (tending toward being OOD), and not completely different.
> >
> > Another idea to pick the threshold is to consider the null distribution of the statistic S(x). That is, consider the distribution of S(x) evaluated on in-sample examples. From this (empirical) distribution (or histogram), one can choose a an appropriate quantile (say 95%) as the threshold. This is exactly the idea from frequentist hypothesis testing. But you did not consider this approach in Section 3. For instance, this idea is considered in
> >
> > E. Nalisnick, A. Matsukawa, Y. W. Teh, and B. Lakshminarayanan. Detecting out-of-distribution inputs to deep generative models using a test for typicality. ArXiv, 1906.02994, 2019b.
> >
> > in your citation list.

---

> > > ### Author Response · Authors · 2019-11-14
> > > **Clarifying C2**
> > >
> > > Thanks for asking again. Yes, the first approach that we mention in the answer is to consider the “null distribution of the statistic S(x)”. We’re sorry if that wasn’t clear. We have now added Section 3.3, which explicitly discusses four possible ways to employ S(x) in practice (the two cases mentioned in our previous answer plus two additional ones). In the same section, we also justify our decision of not using a thresholded variant in order to perform a generic evaluation, and also to compare with other results from the state-of-the-art, which mostly use AUROC. We thank the Reviewer for insisting on this issue.

---

### Official Review · AnonReviewer3 · 2019-10-24
**Official Blind Review #3**

**Rating:** 6

**Review:**

This paper analyzes the peculiar case that deep generative models often assign a higher likelihood to other datasets than they were trained on. The running hypothesis here is, that input complexity plays a central role. Measuring a proxy for input complexity shows that it is tightly anticorrelated with likelihood and therefore seems to describe the trend well. A new OOD detection score is introduced based on these insights.

The failure of Cifar-10 generative models to detect SVHN as OOD via thresholded likelihood is an interesting question. I also think that the conclusion of this work, that this behavior may be natural and a consequence of likelihood itself, seems very sensible.

The overall message of the paper is very important and might help to avoid researchers wasting time on improving likelihood models for OOD detection. Instead, it seems a more sensible way to deal with this problem is to consider likelihood ratios.

One concern is that a formal discussion if the likelihood is supposed to behave the way the authors describe, or if a better model might solve this eventually. I think it is already mentioned that empirically there is no reason to believe a better model would solve this, it would just be nice to have a theoretical statement here as well.

Another concern is that even though the empirical results look quite promising, it would be good to stress-test the proposed score on more common OOD detection benchmarks against state-of-the-art methods to see if likelihood generative models with the proposed criterion are competitive there.

All in all, I think this is a great paper, additional theoretical analysis and stronger empirical results would be helpful to increase the significance of the work.

--------------
Post Rebuttal
--------------

I would like to thank the authors for their interesting response.
However, I will not change my score as the response did not change my opinion on the need for a more theoretical treatment of the statement, which is a serious (and potentially hard to resolve) shortcoming of the paper in my opinion.

**Experience Assessment:**

I have published one or two papers in this area.

**Review Assessment: Checking Correctness Of Derivations And Theory:**

I carefully checked the derivations and theory.

**Review Assessment: Checking Correctness Of Experiments:**

I carefully checked the experiments.

**Review Assessment: Thoroughness In Paper Reading:**

I read the paper thoroughly.

---

> ### Author Response · Authors · 2019-11-08
> **Response to Review #3**
>
> We thank Reviewer 3 for his/her positive assessment of our work. We now comment on the two concerns expressed by the Reviewer (C1-C2).
>
> C1: Better models — The question of whether a better model might solve the problem is an interesting one and our empirical findings suggest a mixed answer. On the one hand, we do not observe strong differences between PixelCNN++ and Glow for OOD detection. Overall, both models yield comparable accuracies, despite PixelCNN++ having much better log-likelihoods and slightly better generative capabilities than Glow (note that the latter is not an issue with our implementation, but a fact well reported in the literature; see for instance [1]). On the other hand, we do observe some differences with increasing model sizes/capabilities in the case of Glow. In that case, larger models tend to improve performance. Thanks to the Reviewers’ suggestions, we have now added this last experiment to Appendix D and comment about it in the main paper. Regarding a more theoretical analysis of the statement, we agree with the Reviewer. However, we believe such analysis deserves a proper, formal, and extensive treatment, which is beyond the scope of the current paper.
>
> C2: Common OOD detection benchmarks — The question whether likelihood-based generative models are competitive with other state-of-the-art approaches can be assessed by looking at the results of Table 2, where the latter are grouped under the term “classifier-based approaches”. We see that both generative- and classifier-based approaches reach similar accuracies, and that best scores do not always correspond to the same single approach. Interestingly, the proposed solution, albeit simple, achieves some of those best scores. Regarding the use of more OOD detection benchmarks, we find that current practice in this task is not already standardized, with no general consensus on a comprehensive pool of benchmarks. At the moment, results for different approaches are scattered across a number of benchmarks, of which we consider the ones that, to our understanding, are the most common ones.
>
> Thank you again for your positive assessment of our work.
>
> [1] J. Ho, X. Chen, A. Srinivas, Y. Duan, P. Abbeel. “Flow++: improving flow-based generative models with variational dequantization and architecture design”. Proc. of ICML, PMLR 97: 2722-2730, 2019.

---

### Decision · Program_Chairs · 2019-12-19

**Decision:**

Accept (Poster)

**Comment:**

I have read the paper and the reviews carefully. Despite the numerical scores, I think this paper is above the bar for ICLR, and recommend acceptance.

This paper addresses the now-well-known problem that generative models often assign higher likelihoods to out-of-distribution examples, rendering likelihoods useless for OOD detection. They diagnose this as resulting from differences in compressibility of the input, and propose to compensate for this by comparing the log-likelihood to the description length from a strong image compressor. They show this performs well against a variety of OOD detection methods.

The idea is a natural one, and certainly should have been one of the first things tried in addressing this phenomenon. I'm a little surprised it hasn't been done before, but none of the reviewers or I are aware of a prior reference, so AFAIK it's novel. One reviewer believes the contribution is small; while it's simple, I think the field will benefit from a careful implementation and testing of this approach.

Multiple reviewers raise the concern of whether generative models' bias towards low-complexity inputs is just a matter of needing better generative models. I don't think so: even arbitrarily good generative models will still be limited by the inherent compressibility of an input (e.g. as measured by Kolmogorov complexity).

I'm also not concerned about the lack of an explicit threshold; if one has proposed a good score function, there are many ways one could choose a threshold, depending on the task.